# Caspase-1 Variants and Plasma IL-1β in Patients with *Leishmania guyanensis* Cutaneous Leishmaniasis in the Amazonas

**DOI:** 10.3390/ijms252212438

**Published:** 2024-11-19

**Authors:** Josué Lacerda de Souza, Marcus Vinitius de Farias Guerra, Tirza Gabrielle Ramos de Mesquita, José do Espírito Santo Junior, Hector David Graterol Sequera, Lener Santos da Silva, Larissa Almeida da Silva, Filipe Menezes Moura, Lizandra Stephanny Fernandes Menescal, Júlia da Costa Torres, Suzana Kanawati Pinheiro, Herllon Karllos Athaydes Kerr, Mauricio Morishi Ogusku, Mara Lúcia Gomes de Souza, Jose Pereira de Moura Neto, Aya Sadahiro, Rajendranath Ramasawmy

**Affiliations:** 1Programa de Pós-Graduação em Biodiversidade e Biotecnologia da Amazonia Legal (Rede Bionorte), Universidade do Estado do Amazonas, Manaus 69055038, Amazonas, Brazil; jzlacerda94@gmail.com (J.L.d.S.); lener.santos77@gmail.com (L.S.d.S.); 2Faculdade de Medicina, Universidade Nilton Lins, Manaus 69058030, Amazonas, Brazil; sdjunior.biol@gmail.com; 3Fundação de Medicina Tropical Doutor Heitor Vieira Dourado, Manaus 69040000, Amazonas, Brazillariadsb@gmail.com (L.A.d.S.); juliacostatorres3@gmail.com (J.d.C.T.); maralgsouza@gmail.com (M.L.G.d.S.); 4Programa de Pós-Graduação em Medicina Tropical, Universidade do Estado do Amazonas, Manaus 69055038, Amazonas, Brazil; tirzagabi@gmail.com (T.G.R.d.M.); hector.graterol4@gmail.com (H.D.G.S.); filipe.moura14@yahoo.com.br (F.M.M.); skanawatip@gmail.com (S.K.P.); karllosathaydesss@gmail.com (H.K.A.K.); 5Programa de Pós-Graduação em Imunologia Básica e Aplicada, Universidade Federal do Amazonas, Manaus 69060001, Amazonas, Brazil; lizandramenescal@gmail.com (L.S.F.M.); jpmn@ufam.edu.br (J.P.d.M.N.); asadahiro@ufam.edu.br (A.S.); 6Laboratório de Micobacteriologia, Instituto Nacional de Pesquisas da Amazônia, Manaus 69060001, Amazonas, Brazil; mmogusku.inpa@uol.com.br; 7Genomic Health Surveillance Network: Optimization of Assistance and Research in The State of Amazonas—REGESAM, Manaus 69055038, Amazonas, Brazil

**Keywords:** *Leishmania guyanensis*, Amazonas, caspase-1, genetic, IL-1β, NLRP3

## Abstract

Leishmaniasis, a disease caused by protozoan *Leishmania* spp., exhibits a broad range of clinical manifestations. Host resistance or susceptibility to infections is often influenced by the genetic make-up associated with natural immunity. Caspase-1, a key component of the NLRP3 inflammasome, is critical for processing pro-IL-1β into its active form, IL-1β, while CARD8 functions as an NLRP3 inflammasome inhibitor. We conducted a case–control study comparing *L. guyanensis*-cutaneous leishmaniasis (*Lg*-CL) patients with healthy individuals (HCs) by analyzing the *CASP1* genetic variants rs530537A>G, rs531542C>T, rs531604A>T and rs560880G>T. Additionally, a combined analysis of *CARD8*rs2043211A>T with *CASP1*rs530537 was performed. The genotype distribution for the four variants showed no significant differences between *Lg*-CL patients and HCs. However, the haplotype analysis of the four *CASP1* variants identified the GTTT haplotype as associated with a 19% decreased likelihood of *Lg*-CL development, suggesting a protective effect against disease progression. The combined analysis of *CARD8* with *CASP1* variants indicated that individuals homozygous for both variants (GG/TT) exhibited a 38% reduced risk of developing *Lg*-CL (OR = 0.62 [95%CI:0.46–0.83]) in comparison to individuals with other genotype combinations. No correlation was found between the CASP1 variant genotypes and plasma IL-1β levels. *CASP1* may act as a genetic modifier in *Lg*-CL.

## 1. Introduction

Leishmaniasis, a neglected vector-borne disease caused by the intracellular protozoan parasites *Leishmania* spp., remains a significant public health issue in the Americas, East Africa, North Africa and West and South-East Asia. According to the WHO, 99 countries were considered endemic for leishmaniasis in 2022, and, among these, 90 were endemic for cutaneous leishmaniasis (CL), of which 71 were also endemic for visceral leishmaniasis (VL) [1]. In 2022, 205,986 cases of CL and 12,842 cases of VL were reported [1]. In 2020, 16,813 new cases of tegumentary leishmaniasis (TL) were reported in Brazil [2], of which 1690 were from the state of Amazonas [2].

Leishmaniasis is transmitted by phlebotomine sandflies during blood meals, introducing *Leishmania* promastigotes into the host. The clinical manifestations of leishmaniasis range from asymptomatic cases and self-healing or non-healing CL to severe mucosal involvement (ML) or life-threatening VL. The host’s clinical response to *Leishmania* infection is closely linked to their immunological response. Disease outcomes depend on the infecting *Leishmania* species, intrinsic virulence factors and the genetic and immune profile of the host [3,4]. More than 20 *Leishmania* species can cause leishmaniasis. In Brazil, the most common species causing CL are *L. braziliensis*, *L. guyanensis*, *L. lainsoni*, *L. naiffi* and *L. lindenbergi.*

The nucleotide-binding domain and leucine-rich repeat-containing receptor protein (NLRP) family comprises a group of intracellular pathogen recognition receptors with diverse functions in regulating innate immunity and inflammation. The activation of NLRP1, NLRP3 and NLRC4 triggers the assembly of inflammasome complexes through the association of apoptosis-associated speck-like protein containing a caspase-1 recruiting domain (ASC) and pro-CASP-1 via Pyrin–Pyrin and CARD–CARD homotypic interactions, respectively. Upon activation, pro-CASP-1 is converted to CASP-1, which processes pro-IL-1β and pro-IL-18 into their active secreted forms, IL-1β and IL-18 [5,6]. Caspase recruitment domain-containing family member 8 (CARD8) acts as a negative regulator of the NLRP3 inflammasome [7]. The FIIND domain of CARD8 interacts with the nucleotide-binding domain (NOD) of NLRP3, while the C-terminal CARD domain interacts with CASP-1 via CARD–CARD homophilic interactions [8,9]. Silencing CARD8 expression through small interfering RNA (siRNA) increases the production of IL-1β [7]. CARD8 also inhibits nuclear factor kappa B (NFkB) activity, leading to the reduced expression of IL-1 β and TNFα [10,11].

The NLRP3 inflammasome is activated during *Leishmania* infection [12,13]. The activation of the NLRP3 inflammasome requires a priming signal to activate the transcription factor nuclear factor kappa B (NF-kB) and the extracellular signal-related kinase (ERK) pathways for the transcription of NLRP3, pro-IL-1β and pro-IL-18. *Leishmania* lipophosphoglycan, glycosphingophospholipids and DNA are recognized by Toll-like receptors (TLR2, TLR4 and TLR9, respectively). During *Leishmania* infection, these TLRs promote the transcription of inflammatory cytokines such as TNFα, IL-6, IL-1β and IL-12. This signaling may function as a primary trigger for NLRP3 inflammasome assembly by facilitating the transcription of NLRP3 via NFκB [12].

In studies with C57BL6 mice deficient in *NLRP3*, *ASC* or *CASP1*, an increase in the lesion size and parasite burden was observed upon infection with *L. amazonensis* [14]. In vitro analyses using macrophages infected with *L. braziliensis*, *L. amazonensis* or *L. mexicana* demonstrated that the NLRP3 inflammasome’s activation is required for IL-1β secretion and host resistance [14], while these mice exhibited protection against the *L. major* Seidman strain, which induces chronic lesions in C57BL6 mice [15]. In contrast, susceptible BALB/c mice showed resistance to *L. major* infection in the absence of the NLRP3 inflammasome components [16]. Similarly, mice lacking *CASP1* or *NLRP3* exhibited reduced pathology when infected with *L. braziliensis* [17].

The double-stranded *Leishmania* RNA virus (LRV) found in *L. guyanensis* inhibits CASP-1 activation and IL-1β production by inducing type-1 interferon signaling via TLR3, leading to the autophagy-mediated degradation of NLRP3 [18]. Bone marrow-derived macrophages (BMDMs) infected with LRV-positive *L. guyanensis* displayed reduced levels of CASP-1 and IL-1β compared to those infected with LRV-negative *L. guyanensis* in vitro. C57BL6 mice infected with LRV-positive *L. guyanensis* exhibited more pronounced ear swelling and higher parasite burdens than those infected with LRV-negative *L. guyanensis*. This difference was diminished in NLRP3-deficient C57BL6 mice infected with LRV-negative *L. guyanensis*, suggesting that NLRP3 and IL-1β play protective roles against *L. guyanensis* infection [18].

Apoptotic bodies released from dying cells can act as danger-associated molecular patterns (DAMPs), activating inflammasomes to trigger CASP-1 activation for the processing of pro-IL-1β [19]. CASP-1 has been shown to be critical in inducing CD8^+^T cell-mediated pathology but not in controlling parasite multiplication in C57BL/6 mice co-infected with *L. major* and lymphocytic choriomeningitis virus (LCMV) [20]. Similarly, wild-type C57BL/6 mice co-infected with *L. major* and LCMV developed severe lesions compared to *NLRP3*-deficient C57BL/6 mice co-infected under the same conditions [20]. CD8+ T cells have been shown to promote inflammation through granule-mediated cytotoxicity in *L. braziliensis* patients [21,22,23,24,25].

Given the dual role of NLRP3 inflammasome activation in leishmaniasis, where it may be protective or pathogenic depending on the infecting *Leishmania* species, and the levels of pro-IL-1β processed into active IL-1β by CASP-1, we investigated whether genetic variants of *CASP1* influenced the plasma levels of IL-1 β in patients with CL caused by *L. guyanensis* and healthy individuals with no history of leishmaniasis from the same endemic region.

## 2. Results

### 2.1. Basic Characteristics of the Study Population

The basic characteristics of the study population have been previously described [26]. Briefly, the study comprised 850 patients with *Lg-*CL and 891 HCs. Among the *Lg*-CL patients, 639 were male, with a mean age of (mean ± standard deviation) 34.4 ± 13.7 years, while the remaining 211 female participants had a mean age of 37.5 ± 15.7 years. In the HC group, there were 608 male (42 ± 17.5 years) and 283 female (40 ± 17.4 years) participants. The male HCs were significantly older than the male *Lg*-CL patients (*p* < 0.0001).

### 2.2. Genotype Frequencies and Statistical Comparison of the Genotypes Between Patients with L. guyanensis and Healthy Controls

This case–control study compared unrelated *Lg*-CL patients to unrelated healthy individuals, following the Strengthening the Reporting of Genetic Association Studies (STREGA) guidelines. The *CASP1* gene is located on chromosome 11q22.3 and comprises nine exons, as shown in Figure 1A. The variants rs530537A>G, rs531542C>T, rs531604A>T and rs560880G>T are located in intron 8. The genotype frequencies of the *CASP1* variants (rs530537A>G; rs531542C>T; rs531604A>T; rs560880G>T) are presented in Table 1. All four variants studied conformed to the Hardy–Weinberg equilibrium expectation. Based on the success rates of direct nucleotide sequencing, 780, 745, 745 and 725 *Lg*-CL patients had high-quality genotypic data for the variants rs530537, rs531542, rs531604 and rs560880, respectively. For the HC group, the corresponding success rates were 760, 770, 765 and 752, respectively. The genotype distribution for the four variants did not significantly differ between *Lg*-CL patients and HCs.

Figure 1B displays the linkage disequilibrium (LD) between the four variants in Lg-CL patients, the HCs and all study participants. The degree of pairwise correlation between the nucleotides was very high, as indicated by the D’ values, suggesting that all four variants are in strong LD. Only 717 *Lg*-CL patients and 730 HCs had complete genotypic data for the four variants. Six haplotypes were generated using the Haploview software, as shown in Table 2. The GTTT haplotype is associated with resistance to the progression of *Lg*-CL development. Individuals with the GTTT haplotype have a 19% reduced likelihood of developing CL upon infection with *L. guyanensis*.

### 2.3. Genetic Combination of Variant rs530537 of CASP1 and rs2043211 of CARD8

CARD8, a negative regulator of the NLRP3 inflammasome, inhibits the transcription factor NF-ĸB, leading to the reduced expression of IL-1β and TNFα [9,10]. The variant rs2043211 A>T is a transversion from Adenine to Thymine, resulting in a premature stop codon at position ten (Cys10stop codon) [27]. Recent findings from our study population demonstrate that this variant is not associated with the development of *Lg*-CL [26]. The rs2043211 A>T genotypes (AA, AT and TT) had frequencies of 49%, 42% and 9% among the *Lg*-CL patients, respectively, and 52%, 41% and 7% among the HCs [26]. The T allele of rs2043211 has been suggested to correlate with increased NFĸB activity [27]. We conducted a combined analysis of the *CADR8* variant rs2043211 with *CASP1* rs530537, as shown in Table 3. Individuals homozygous for both variants (GG/TT) demonstrated a 38% reduced risk of developing *Lg*-CL (OR = 0.62 [95%CI: 0.46–0.83]) in comparison to individuals with other combined genotypes.

### 2.4. CASP1 Variants and Plasma IL-1β Levels

CASP-1 is responsible for processing pro-IL-1β into the active form, IL-1β. We investigated whether the genotypes of the *CASP1* variants were associated with differences in the plasma IL-1β levels. Figure 2 illustrates the plasma IL-1β levels across different genotypes. None of the studied variants showed a significant correlation with the plasma IL-1β levels.

We also investigated whether individuals homozygous for genetic combination GG/TT of variants *CASP1* rs530537 and *CARD8* rs2043211 correlated with the plasma IL-1β levels, given that the rs2043211 TT genotype of *CARD8* has been associated with high NF-ĸB activity [27]. As shown in Figure 3A, no significant differences were observed among the combined genotypes, despite the mean plasma IL-1β levels for AG/AT being 1.39 ± SD2.31 compared to AA/TT (0.87 ± 0.87); GG/TT (0.88 ± 1.54); AA/AA (0.94 ± 2.03); and GG/AA (0.74 ± 0.65). Figure 3B compares the plasma IL-1β levels of the GG/TT genotype to those of the other the combined genotypes (REST), showing no significant differences.

## 3. Discussion

The immune response in leishmaniasis has been extensively studied in animal models, particularly in resistant C57BL/6 and susceptible BALB/c mouse strains infected with *L. major*, to understand the cellular and molecular mechanisms underlying the disease. BALB/c mice display a Th2-type immune response to *L. major*, leading to disease development through the production of Th2 cytokines such as IL-4, IL-13 and IL-10, while C57BL/6 mice mount a robust Th1-type response, conferring resistance that depends primarily on the IL-12/IFN-γ/TNF axis [28]. The multiplication of *Leishmania* parasites is controlled by IFN-γ, which activates macrophages to mediate parasite killing.

In humans, progress has been made in the understanding of the cellular and molecular mechanisms driving leishmaniasis progression, but further research is required to understand why only a subset of individuals in endemic areas develop the disease. Understanding the host predisposition to the development of the disease, particularly in immune response genes, can lead to the development of alternative therapies such as immunotherapy, since chemotherapeutic drugs for leishmaniasis are often toxic and may be ineffective, with treatment success ranging from only 53 to 70% [29]. Indeed, efforts are being made to test plant metabolites that increase reactive oxygen species, which are potent inhibitors of *Leishmania* multiplication, in vitro in human cells [30]. Recently, two metabolites, deoxyalpinoid B and sulforaphane, have shown promising results in controlling *L. donovani* and *L. infantum* multiplication in vitro [31].

This study aimed to contribute to and advance our understanding of the genetic mechanisms underlying disease progression, as leishmaniasis remains a major neglected disease, affecting millions of people worldwide, with approximately 2.4 million disability-adjusted life years (DALYs) lost globally due to CL and VL [32]. Leishmaniasis is the second-leading cause of mortality and fourth-leading cause of morbidity among tropical diseases [33]. CL alone accounts for approximately 41,700 DALYs [34]. Currently, there is no effective vaccine or highly efficacious treatment.

Previous studies have shown that leishmaniasis may be influenced by the genetic factors of the host [35]. Here, we did not observe any association of the four *CASP1* variants studied with either susceptibility to or protection against Lg-CL. In Chagas disease, caused by the protozoan *Trypanosoma cruzi*, another *CASP1* variant, rs12417050, located at nucleotide 105065283, was not associated with the disease [36]. However, we showed that a haplotype (GTTT) derived from the four variants located in intron eight of *CASP1* is associated with protection against *Lg*-CL. Furthermore, a combined analysis of the *CARD8* variant rs2043211 with *CASP1* rs530537 revealed that individuals homozygous for both variants (GG/TT) had a 38% reduced risk of developing *Lg*-CL (OR = 0.62 [95%CI: 0.46–0.83]) in comparison to individuals with other combined genotypes. CARD8 has been shown to inhibit the major transcription factor controlling innate immunity and inflammation, NF-ĸB [10,11]. The *CARD8* variant rs2043211 generates a premature stop codon in the presence of the T allele; interestingly, one study has showed that the TT genotype correlates with the high activity of NF-ĸB in vitro [27]. A plausible explanation is that the loss of inhibition of NF-ĸB activity by CARD8 may provide an advantage in a subset of individuals carrying the combined genotypes GG/TT, generating sufficient proinflammatory cytokines to control the infection.

Several studies have cited a deleterious effect of NLRP3 [15,16,17], while others have indicated a protective role during *Leishmania* infection [14]. The activation of the NLRP3 inflammasome promotes the activation of CASP-1 to process pro-IL-1β into active IL-1β. The signaling pathway of IL-1β, a strong pro-inflammatory cytokine, subsequently induces more inflammation [37]. Recently, we have shown that *IL1B* variant genotypes –511T/C (rs16944) and +3954C/T (rs1143634) did not correlate with the plasma IL-1β levels [38]. In animal models, *CASP1* knock-out mice have significantly lower levels of IL-1β compared to wild-type mice [39,40]. The reduced expression of the inflammasome adaptor molecule ASC has been shown to limit the proteolytic processing of CASP-1 in murine hepatocytes, leading to low levels of IL-1β and promoting *Plasmodium* infection [41]. IL-1β upregulates the potent neutrophil chemoattractants CXCL1, CXCL2 and CCL4 [42]. Neutrophils have been shown to promote pathogenesis in murine *Leishmania* infection [43], while neutropenic mice are resistant to the *L. major* Seidman strain [15].

In light of these studies, IL-1β regulation must be finely tuned to eliminate the pathogen without contributing to immunopathogenesis. However, in LRV-positive *Lg*-CL patients, sufficiently high plasma IL-1β levels may provide an advantage, as the virus inhibits CASP-1 to contain parasite replication. In the current study, the plasma IL-1β levels did not correlate with any genotype of the four *CASP1* variants studied. Further studies should explore additional variants in the *CASP1* gene to better understand the impact of the genetic variation on the plasma IL-1β levels. Notably, other genes in the inflammasome complex are also critical to study, as IL-1β regulation does not depend solely on *CASP1*. Additionally, a comprehensive genetic analysis of all inflammasome components is essential to elucidate the biological significance of these variants. Indeed, only individuals heterozygous for both variants of *CASP1* rs530537 and *CARD8* rs2043211 tended to show elevated plasma IL-1β levels.

Our study was limited by the absence of screening for the LRV virus to stratify the patients into LRV-positive and LRV-negative groups, as well as the lack of delayed-type hypersensitivity skin testing with *Leishmania* antigens in the control group to confirm parasite exposure. However, the LD structure was similar between the patients and HCs, indicating that both groups were drawn from the same population. Another limitation was the lack of a plasma IL-18 assay.

Resistance or susceptibility to pathogens often depends on the genetic make-up of the host in terms of natural immunity. Here, we show that a *CASP1* haplotype, as well as combined homozygous *CASP1* and *CARD8* genotype variants, are associated with protection against the development of Lg-CL.

## 4. Materials and Methods

### 4.1. Area of the Study Population

This study was carried out in the surrounding regions of the city of Manaus, the capital of the state of Amazonas, Brazil, near the BR-174 and AM-010. Over the years, there has been significant deforestation due to settlement expansion, agriculture and livestock farming. These activities have transformed these areas into endemic zones for *L. guyanensis* infection, particularly in the communities of Pau-Rosa, Cooperativa, Água-Branca, Leao, and Brasileirinho. The state of Amazonas is situated in the extreme north of Brazil at latitude −19,9657 and longitude −44,0413. Patients with active CL (presenting with ≤six lesions), affected by *L. guyanensis*, were recruited between November 2012 and April 2017 at the Fundação de Medicina Tropical Dr. Heitor Vieira Dourado (FMT-HVD), a regional reference center for tropical diseases. The control group consisted of healthy subjects (HCs) from the same endemic area as the patients. These healthy subjects had no history of leishmaniasis and most worked in agriculture, similarly to the patients. A considerable proportion of the study participants reported a history of malaria, as these regions also are endemic for *Plasmodium vivax* malaria. This population represents an admixture group of Native American ancestry, commonly referred to as caboclo, comprising 50–60% Native American, 40–50% European and approximately 10% African ancestry [44]. This study involved 1714 unrelated individuals (855 patients with Lg-CL and 859 HCs), as previously described [38,45,46,47]. Patients were diagnosed with CL for the first time and presented with six or fewer lesions, with most having a single lesion. All patients had active CL. HCs were not stratified as asymptomatic, as we did not perform the delayed hypersensitivity test for Leishmania antigens. All participants tested negative for HIV and had no history of cardiac, renal or diabetes diseases. These HCs underwent a thorough physical examination by physicians to rule out any doubts regarding their previous history of leishmaniasis. They shared similar socio-epidemiological conditions.

### 4.2. Sample Size Calculations

The effective sample size was determined using the Genetic Power calculator of Harvard University (https://zzz.bwh.harvard.edu/gpc/ accessed in 2 November 2014), considering multiple gene inputs for a trait. The calculation was based on several assumptions, including a minor allele frequency of 5%, disease prevalence of 5%, complete linkage disequilibrium between the marker and the trait, a case–control ratio of 1, a type 1 error rate of 5% and an odds ratio of 1.5 and 2.0 for heterozygotes and homozygotes, respectively, with power of 80%. The genetic allelic model indicated a required sample size of 789 cases and 789 controls.

### 4.3. Ethical Statement

This study adhered to the principles outlined in the Declaration of Helsinki and received approval from the Research Ethics Committee of the Fundação de Medicina Tropical Dr. Heitor Vieira Dourado under the file number CAAE:09995212.0.0000.0005. Written informed consent was obtained from all participants for biological sample collection and subsequent analysis. For participants under 18 years of age, written informed consent was obtained from a parent or guardian.

### 4.4. Identification of Leishmania Species

All CL patients who had microscopic confirmation of *Leishmania* parasites in Giemsa-stained lesion scarifications provided biopsy specimens from their skin ulcer lesions for DNA extraction to identify the *Leishmania* species. Species identification was performed through the direct nucleotide sequencing of a fragment of the HSP70 gene (233bp) and mini exon gene (227 bp) amplified by polymerase chain reaction (PCR). PCR products were purified using 20% PEG, following previously described protocols [38]. Sequencing was performed using an ABI 3130XL automated DNA sequencer (Applied Biosystem, Life technologies, Waltham, MA, USA) with the POP-7 polymer. Nucleotide readings were carried out using the Sequencing Analysis software (Applied Biosystems, v5.3.1). The sequencing was conducted using the BigDye Terminator v3.1 Cycle Sequencing Kit (Thermo Fisher, Waltham, MA, USA), following the manufacturer’s protocols.

### 4.5. Collection of Biological Samples and DNA Isolation

For genomic DNA isolation using the protein K salting-out method [48] and for the measurement of circulating plasma cytokines, each participant provided five milliliters of peripheral blood, collected into EDTA-containing Vacutainers (Becton Dickinson, Sao Paulo, Brazil) through venipuncture.

### 4.6. Genotyping of CASP-1 Variants

A pair of primers (CASP1F 5′-CTCTGTATGTAATGACAGACAC-3′ and CASP1R 5′-CCTTACTCCTACCTTCTAGATG-3′) was designed to flank four single-nucleotide variants (SNVs) (rs530537; rs531542; rs531604; rs560880) located in intron 8 of the CASP-1 gene. The forward and reverse primers spanned nucleotide positions 105,027,602 to 105,027,631 and 105,027,930 to 105,027,951, respectively, on chromosome 11q22.3 build Grch38p14, respectively, generating a 349bp fragment. PCR reactions were performed in the Applied Biosystem Veriti Thermal Cycler (Life technologies, Waltham, MA, USA), containing 50 ng of genomic DNA in a final volume of 25 μL with 1.5 mmol/L of MgCl_2_, 0.25 pmol/L of the forward and reverse primers, 40 μmol/L of each dNTP and 1U of Taq polymerase in buffer containing 100 mmol/L of Tris–HCL (pH 8.3) and 500 mmol/L KCL. The PCR cycling conditions consisted of initial denaturation at 95 °C for 5 min, followed by 35 cycles at 95 °C for 15 s, 15 s at 58 °C, 30 s at 72 °C and a final extension step at 72 °C for 7 min.

The 349bp amplicons were sequenced using either the forward (CASP1F) or reverse (CASP1R) primers with the BigDye Terminator v3.1 Cycle Sequencing Kit (Thermo Fisher, Waltham, MA, USA), according to the manufacturer’s protocol, in the Applied Biosystem Veriti Thermal Cycler, as described above [38]. Only high-quality sequences were used for variant analysis.

### 4.7. Measurement of Plasma IL-1β Levels by Luminex

Plasma samples were stored at −80 °C until cytokine level measurement. The collection of plasma from patients and controls for cytokine analysis took place between January 2013 and March 2016 and included 354 patients with *Lg-*CL (264 males and 90 females) and 376 (269 males and 107 females) healthy individuals. Plasma IL-1β levels were measured in 5 µL of plasma with the Human Cytokine Grp I Panel 27-Plex kit (Bio-Rad, Hercules, California. USA) through a multiplex cytokine assay on the Bio-plex 200 Protein Array System (Luminex Corporation, Austin, Texas, USA).

### 4.8. Statistical Analyses

The statistical comparison of the genotypes between patients with *Lg*-CL and HCs was performed using logistic regression analysis, calculating the odd ratio (OR) and 95% confidence interval (CI), with adjustments for sex and age, using the R software version 4.3.1 of the SNPassoc package 2.1-0, which also supports different inheritance models (codominant, dominant, recessive and overdominant). Allele frequencies were calculated using the gene counting method, and the Hardy–Weinberg equilibrium was assessed by comparing the observed and expected genotype frequencies. Linkage disequilibrium analysis, haplotype construction and comparisons were conducted using the Haploview software 4.2. The plasma IL-1β levels by genotype were analyzed using a Generalized Linear Model (GLM) for quantitative traits in the R software with the SNPassoc package and visualized using the ggplot2 package. Post hoc analysis following ANOVA was conducted using the postHoc package version 0.1.3 in the R software (www.r-project.org) to compare the effect levels based on the genotypes, and the *p*-values were corrected using the false discovery rate (FDR).

## Figures and Tables

**Figure 1 ijms-25-12438-f001:**
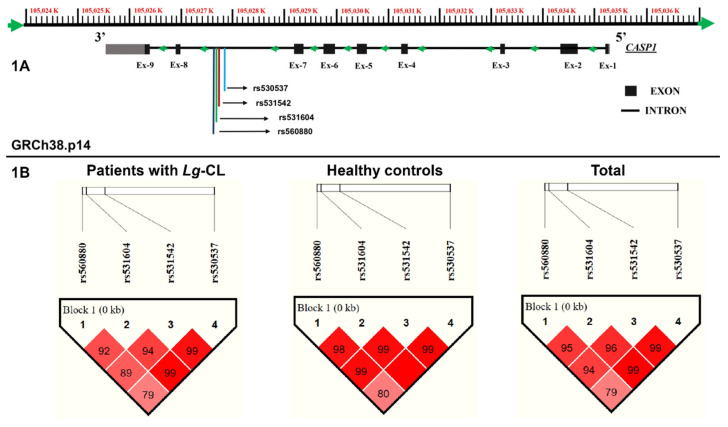
Structure of the *CASP1* gene as adapted from the NCBI (Build GRCh38.p14) showing the localization of the four variants in (**A**) and linkage disequilibrium between the variants among patients with cutaneous leishmaniasis (cases), healthy controls and all study participants (total) in (**B**).

**Figure 2 ijms-25-12438-f002:**
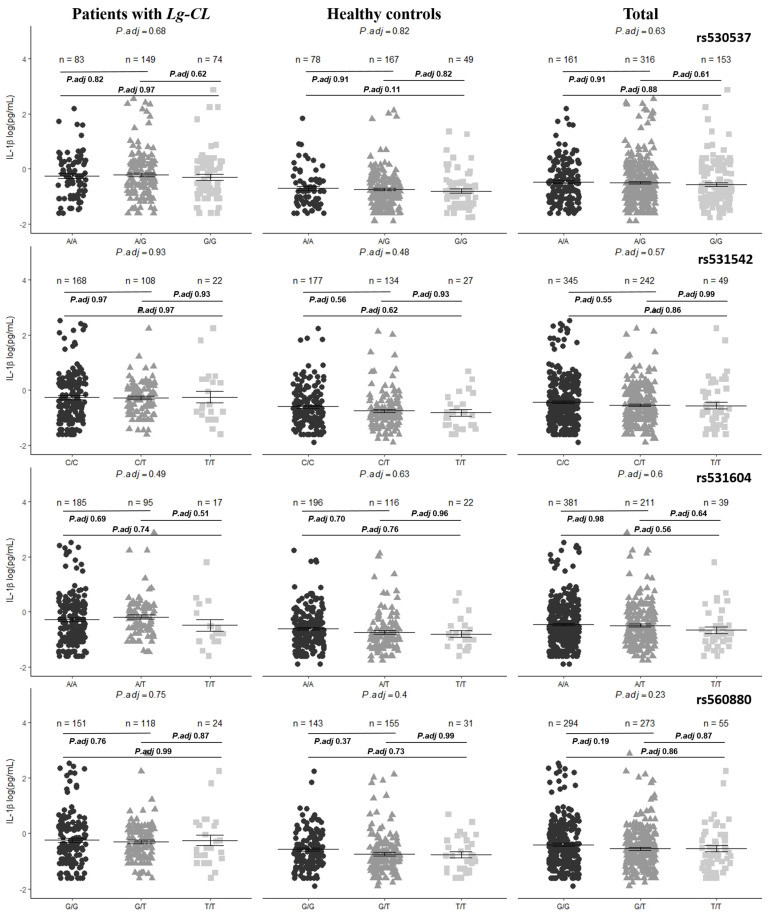
Plasma IL-1β levels by genotype (circle—common homozygote, triangle—heterozygote and square—rare homozygote) for the four variants (rs530537, rs531542, rs531604 and rs560880) in patients with *Leishmania guyanensis* cutaneous leishmaniasis, healthy controls and all participants in the cytokine assay.

**Figure 3 ijms-25-12438-f003:**
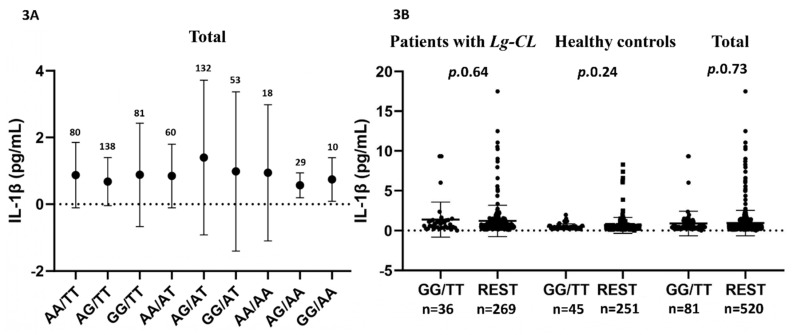
Plasma IL-1β levels for combined genotypes of variants rs530537 of *CASP1* and rs2043211of *CARD8* in all participants with cytokine assay, including patients with *Leishmania guyanensis* cutaneous leishmaniasis and healthy controls (total) (**A**) and the GG/TT to other combined genotypes (REST) (**B**).

**Table 1 ijms-25-12438-t001:** Distribution of genotypes and statistical comparison of the variants of *CASP1* between patients with *Leishmania guyanensis* cutaneous leishmaniasis and healthy controls.

Genotype Frequencies			Statistical Comparison of the Genotypes			
	Cases (%)	HC (%)	Comparison	OR [95% CI]	*p*-Value	AIC
**rs530537**	*n* = 780	*n* = 760	AA vs. AG	1.14 [0.89–1.47]	0.12	2072
AA	215 (27.6)	179 (23.6)	AA vs. GG	1.35 [1.01–1.80]		
AG	384 (49.2)	380 (50)	AA vs. AG + GG	1.21 [0.96–1.53]	0.11	2071
GG	181 (23.2)	201 (26.4)	AA + AG vs. GG	1.23 [0.97–1.56]	0.08	2071
			AA + GG vs. AG	1.00 [0.81–1.21]	0.91	2074
**rs531542**	*n* = 745	*n* = 770				
CC	426 (57.2)	412 (53.5)	CC vs. CT	1.10 [0.88–1.37]	0.20	2034
CT	267 (35.8)	289 (37.5)	CC vs. TT	1.41 [0.95–2.10]		
TT	52 (7)	69 (9)	CC vs. CT + TT	1.15 [0.93–1.41]	0.19	2033
			CC + CT vs. TT	1.36 [0.93–2.00]	0.11	2032
			CC + TT vs. CT	1.05 [0.85–1.35	0.65	2035
**rs531604**	*n* = 745	*n* = 765				
AA	462 (62)	446 (58.3)	AA vs. AT	1.17 [0.94–1.46]	0.25	2027
AT	238 (32)	264 (34.5)	AA vs. TT	1.28 [0.84–1.96]		
TT	45 (6)	55 (7.2)	AA vs. AT + TT	1.19 [0.96–1.47]	0.11	2025
			AA + AT vs. TT	1.21 [0.80–1.84]	0.36	2027
			AA + TT vs. AT	1.14 [0.92–1.42]	0.23	2026
**rs560880**	*n* = 725	*n* = 752				
GG	359 (49.5)	346 (46.1)	GG vs. GT	1.06 [0.85–2.01]	0.18	1982
GT	301 (41.5)	318 (42.3)	GG vs. TT	1.40 [0.98–2.01]		
TT	65 (9)	87 (11.6)	GG vs. GT + TT	1.12 [0.91–1.39]	0.27	1982
			GG + GT vs. TT	1.36 [0.96–1.93]	0.07	1980
			GG + TT vs. GT	1.00 [0.81–1.24]	0.98	1983

Abbreviations: cases, patients with *Leishmania guyanensis* cutaneous leishmaniasis; HC, healthy controls; OR, odds ratio; CI, confidence interval; AIC, Akaike Information Criterion.

**Table 2 ijms-25-12438-t002:** Distribution of haplotypes in patients with *Leishmania guyanensis* cutaneous leishmaniasis and healthy controls as derived by Haploview software 4.2.

Haps	SNV 1	SNV 2	SNV 3	SNV 4	Cases *n* 1434 (%)	HC *n* 1460 (%)	X^2^	OR	CI 95%	*p*-Value
1	A	C	A	G	47,5	44,5	2.4	1.12	0.97–1.30	0.11
2	G	C	A	G	22,6	23	0.002	0.99	0.83–1.18	0.96
3	G	T	T	T	20,8	24,5	5.5	0.81	0.67–0.96	0.01
4	G	T	A	T	3,77	3,2	0.003	1.01	0.69–1.48	0.95
5	A	C	A	T	2,38	3,1	1.37	0.76	0.48–1.19	0.24
6	G	C	A	T	1,18	1,7	1.08	0.71	0.38–1.30	0.29

Abbreviations: SNV, single-nucleotide variant; 1, rs530537 A>G; 2, rs531542 C>T; 3, rs531504 A>T and n4 rs560880 G>T according to the position in the IL13 gene; χ2, chi-squared test; OR, odds ratio; CI, confidence interval.

**Table 3 ijms-25-12438-t003:** Genetic combination of variants rs530537 of *CASP1* and rs2043211 of *CARD8* in patients with *Leishmania guyanensis* cutaneous leishmaniasis and healthy controls.

Combined Genotypes	Patients with Lg-CL *n* = 762 (%)	Healthy Controls *n* = 702 (%)	Total *n* = 1464 (%)
**Rs530537/rs2043211**			
AA/TT	106 (14)	81 (11.5)	187 (13)
AG/TT	185 (24)	160 (23)	345 (23.5)
GG/TT	86 (11.2)	120 (17)	206 (14)
AA/AT	83 (10.8)	73 (10.3)	156 (10.5)
AG/AT	163 (21.5)	152 (21.6)	315 (21.5)
GG/AT	73 (9.5)	60 (8.5)	133 (9)
AA/AA	19 (2.5)	16 (2.3)	35 (2.5)
AG/AA	29 (4)	31 (4.5)	60 (4)
GG/AA	18 (2.5)	9 (1.3)	27 (2)

## Data Availability

All data are included in the manuscript.

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
