# Peer review of "Caspase-1 Variants and Plasma IL-1β in Patients with Leishmania guyanensis Cutaneous Leishmaniasis in the Amazonas"

_ijms, 2024, doi:10.3390/ijms252212438_

Round 1
Reviewer 1 Report
Comments and Suggestions for Authors
The manuscript titled “Caspase-1 variants and plasma IL-1β in patients with Leishmania guyanensis cutaneous leishmaniasis in the Amazonas” presents a case-control study investigating the association between genetic variants of the CASP1 and CARD8 genes and susceptibility to cutaneous leishmaniasis (Lg-CL) caused by Leishmania guyanensis. The authors analyzed specific single nucleotide polymorphisms (SNPs) in the CASP1 gene and their interaction with CARD8 variants, assessing the impact on disease susceptibility and plasma IL-1β levels. This article also explores haplotypes and gene-gene interactions, providing a more nuanced understanding of the genetic factors involved in Lg-CL susceptibility. The identification of the GTTT haplotype as potentially protective against Lg-CL is a noteworthy finding. The authors adhere to the STREGA guidelines for genetic association studies, which enhances the reliability of the findings. The case-control design is appropriate for assessing genetic associations with disease susceptibility. The study addresses a significant public health issue, as leishmaniasis remains a neglected tropical disease affecting millions globally. The focus on genetic factors influencing susceptibility to Lg-CL is timely and relevant, given the increasing incidence of the disease in endemic regions. The results presented are in agreement with the discussion provided. Overall, this article provides valuable insights into the genetic factors influencing susceptibility to Lg-CL and the reviewer has the following comments to the authors that need to be addressed.
1. The discussion could benefit from a more in-depth exploration of the immunopathogenic mechanisms underlying the observed associations. While the article mentions the dual role of NLRP3, a more detailed analysis of how these genetic variants interact with the immune system in the context of leishmaniasis would enhance the manuscript.
2. The lack of screening for Leishmania RNA virus (LRV) in the study population is a significant limitation, as LRV status could influence the immune response and disease progression. This oversight may confound the results and interpretations regarding the role of CASP1 and CARD8 variants.
3. Diarylheptanoid compounds, such as centrolobine and linear diarylheptanoids, are recognized for their anti-leishmanial activity, particularly against Leishmania, a major health concern in Brazil. Considering their potential therapeutic applications, the reviewer recommends that the authors cite relevant articles in the introduction providing significant insights into anti-leishmanial compounds.
https://aces.onlinelibrary.wiley.com/doi/full/10.1002/asia.202400380
https://chemistry-europe.onlinelibrary.wiley.com/doi/abs/10.1002/ejoc.201300097
4. The study reports no significant correlation between CASP1 variants and plasma IL-1β levels, which may limit the interpretation of the biological relevance of the genetic variants studied. As the lack of correlation raises questions about the functional impact of these variants on the inflammatory response, it may be considered to be included in future studies.
Author Response
- The discussion could benefit from a more in-depth exploration of the immunopathogenic mechanisms underlying the observed associations. While the article mentions the dual role of NLRP3, a more detailed analysis of how these genetic variants interact with the immune system in the context of leishmaniasis would enhance the manuscript.
REPLY
Thank you very much for this suggestion. We have included the following in the discussion section:
CARD 8 has been shown to inhibit the major transcription factor controlling innate immunity and inflammation, NF-ĸB [10,11]. The CARD8 variant rs2043211 generates a premature stop codon in the presence of the T allele and interestingly, one study has showed that the TT genotype correlates with high activity of NF-ĸB in vitro [27]. A rationale will be that the lost of inhibition of NF-ĸB activity by CARD8 may be an advantage in a subset of individuals carriers of the combined genotypes GG/TT, generating sufficient proinflammatory cytokines to control the infection.
Please note that we did not want to speculate as there are not many papers talking about these variants.
- The lack of screening for Leishmania RNA virus (LRV) in the study population is a significant limitation, as LRV status could influence the immune response and disease progression. This oversight may confound the results and interpretations regarding the role of CASP1 and CARD8 variants.
REPLY
We agree with you. That is why we focused mainly on the Leishmania guyanensis infection in individuals who developed the disease.
- Diarylheptanoid compounds, such as centrolobine and linear diarylheptanoids, are recognized for their anti-leishmanial activity, particularly against Leishmania, a major health concern in Brazil. Considering their potential therapeutic applications, the reviewer recommends that the authors cite relevant articles in the introduction providing significant insights into anti-leishmanial compounds.
https://aces.onlinelibrary.wiley.com/doi/full/10.1002/asia.202400380
https://chemistry-europe.onlinelibrary.wiley.com/doi/abs/10.1002/ejoc.201300097
REPLY
We have included in the discussion section to make it fluid for the reader
Indeed, efforts are being made by testing plant metabolites that increase reactive oxygen species, potent inhibitor of Leishmania multiplication, in vitro in human cells [30] and recently, two metabolites, deoxyalpinoid B and Sulforaphane have shown promising results in controlling L. donovani and L. infantum multiplications in vitro [31]
- The study reports no significant correlation between CASP1 variants and plasma IL-1β levels, which may limit the interpretation of the biological relevance of the genetic variants studied. As the lack of correlation raises questions about the functional impact of these variants on the inflammatory response, it may be considered to be included in future studies.
REPLY
We agree and have included in the discussion section that future studies should focus on other variants
Other variants in the CASP1 gene should be considered in future studies to understand the impact of the genetic analysis on plasma IL-1β levels. Of note, other genes in the inflammasome complex are also important to study as the regulation of IL-1β does not depend solely on CASP1. Furthermore, combined genetic analysis of all the component of the inflammasome complex is required to provide an understanding of the biologic importance of the variants
Reviewer 2 Report
Comments and Suggestions for Authors The genetic background of an individual, related to inflammasome complex proteins, may affect the susceptibility of Leishmania spp. infection. The manuscript contributes important new information to this field.The two main limitations of the article were highlighted by the authors: screen for LRV virus and confirm parasites exposure in the control group. Some questions need to be answered and suggestions incorporated into the manuscript text. Questions: 1. "Combined analysis of the CARD8 with CASP1 variants indicated that individuals homozygous for both variants (GG/TT) exhibited a 38% reduced risk of developing Lg-CL.". What impact do these variations have on the protein? This has not been discussed. Furthermore, absence of the analysis of gene expression and the determination of serum levels of proteins also limited the study. "Upon activation, pro-CASP-1 is converted to CASP-1, which processes pro-IL-1β and pro-IL-18 into their active secreted forms, IL-1β and IL-18." The authors demonstrated that "No correlation was found between CASP1 variants genotypes and plasma IL-1β levels." 2. Did the authors investigate IL-1β polymorphisms? This information is relevant to associate with serum levels and polymorphisms of other genes. 3. What about IL-18? It is another important factor in the biological process that is the focus of this manuscript. Suggestions: 1. Page 1, Lines 41-42: "a neglected vector-borne disease caused by the intracellular protozoan parasite, Leishmania (L.) spp.," change to Leishmania spp., since the subgenus Viannia can also cause the disease. 2. Page 2, Lines 47-48: "In 2020, 16,813 new cases of tegumentary leishmaniasis (TL) were reported in Brazil 47 [2], of which 1,690 were from the state of Amazonas [2]." The cited reference is from 2018 and does not talk about epidemiology. 3. Page 2, Lines 73-74: "In studies with C57BL6 mice deficient in NLRP3, ASC, or CASP1, an increase in lesion size and parasite burden was observed upon infection with L. amazonensis." This sentence needs references. 4. Page 2, Lines 80-81: "Similarly, mice lacking CASP1 or NLRP3 exhibited reduced pathology when 80 infected with L. braziliensis [15]." Page 2, Lines 82-84: "The double stranded Leishmania RNA virus (LRV) found in L. guyanensis inhibits CASP-1 activation and IL-1β production by inducing type-1 interferon signaling via TLR3, leading to autophagy-mediated degradation of NLRP3 [16]." The cited references do not correspond to the correct articles. 5. The paragraph from lines 92-99 is out of context, I need to be directed to the beginning of the introduction. 6. Page 3, Lines 100-101. "Inflammation in L. braziliensis patients has been associated with granule-mediated cytotoxicity driven by CD8+ T cells [19–23]." The authors talk about patients and then return with in vivo experiments in mice. This sequence should be modified. 7. Page 5, Lines 146-147. "Figure 1B displays the LD between the four variants in Lg-CL patients, the HCs and all study participants." Authors must indicate the meaning of the acronym LD. 8. Page 8, Lines 197 -211. The first two paragraphs of the Discussion need references to the information presented.Author Response
Questions: 1. "Combined analysis of the CARD8 with CASP1 variants indicated that individuals homozygous for both variants (GG/TT) exhibited a 38% reduced risk of developing Lg-CL.". What impact do these variations have on the protein? This has not been discussed.
Thank you for helping us to improve the manuscript
CARD 8 has been shown to inhibit the major transcription factor controlling innate immunity and inflammation, NF-ĸB [10,11]. The CARD8 variant rs2043211 generates a premature stop codon in the presence of the T allele and interestingly, one study has showed that the TT genotype correlates with high activity of NF-ĸB in vitro [27]. A rationale will be that the lost of inhibition of NF-ĸB activity by CARD8 may be an advantage in a subset of individuals carriers of the combined genotypes GG/TT, generating sufficient proinflammatory cytokines to control the infection.
Furthermore, absence of the analysis of gene expression and the determination of serum levels of proteins also limited the study. "Upon activation, pro-CASP-1 is converted to CASP-1, which processes pro-IL-1β and pro-IL-18 into their active secreted forms, IL-1β and IL-18." The authors demonstrated that "No correlation was found between CASP1 variants genotypes and plasma IL-1β levels." 2. Did the authors investigate IL-1β polymorphisms? This information is relevant to associate with serum levels and polymorphisms of other genes.
Recently, we have shown that the genotypes of the IL1B variants –511T/C (rs16944) and +3954C/T (rs1143634) did not correlate with plasma IL-1β levels [47].
- What about IL-18? It is another important factor in the biological process that is the focus of this manuscript.
We included this as a limitation as we did not assay IL-18. Thank you for this.
Suggestions: 1. Page 1, Lines 41-42: "a neglected vector-borne disease caused by the intracellular protozoan parasite, Leishmania (L.) spp.," change to Leishmania spp., since the subgenus Viannia can also cause the disease.
Corrected
- Page 2, Lines 47-48: "In 2020, 16,813 new cases of tegumentary leishmaniasis (TL) were reported in Brazil 47 [2], of which 1,690 were from the state of Amazonas [2].
" The cited reference is from 2018 and does not talk about epidemiology.
Thank you very much for pinpointing this. Actually, the reference was not included in the reference section. We have gone thoroughly in the reference section to make sure they are well cited in the text.
- Page 2, Lines 73-74: "In studies with C57BL6 mice deficient in NLRP3, ASC, or CASP1, an increase in lesion size and parasite burden was observed upon infection with L. amazonensis." This sentence needs references.
Reference included
- Page 2, Lines 80-81: "Similarly, mice lacking CASP1 or NLRP3 exhibited reduced pathology when infected with L. braziliensis [15]."
Corrected
- Page 2, Lines 82-84: "The double stranded Leishmania RNA virus (LRV) found in L. guyanensis inhibits CASP-1 activation and IL-1β production by inducing type-1 interferon signaling via TLR3, leading to autophagy-mediated degradation of NLRP3 [16]." The cited references do not correspond to the correct articles.
Corrected
- The paragraph from lines 92-99 is out of context, I need to be directed to the beginning of the introduction.
Agreed
- Page 3, Lines 100-101. "Inflammation in L. braziliensis patients has been associated with granule-mediated cytotoxicity driven by CD8+ T cells [19–23]." The authors talk about patients and then return with in vivo experiments in mice. This sequence should be modified.
Agreed
- Page 5, Lines 146-147. "Figure 1B displays the LD between the four variants in Lg-CL patients, the HCs and all study participants." Authors must indicate the meaning of the acronym LD.
Corrected
- Page 8, Lines 197 -211. The first two paragraphs of the Discussion need references to the information presented.
References included
Round 2
Reviewer 2 Report
Comments and Suggestions for Authors
The authors made the necessary modifications, and the experiments not performed are indicated as limitations of the study. The current version of the manuscript has been improved.